

# Surface saline lakes in the Mediterranean Sea

Elena Terzić[1], Clara Gardiol[2], and Ivica Vilibić[1,3]

[1]Ruđer Bošković Institute, Bijenička cesta 54, 10000 Zagreb, Croatia
[2]SeaTech, Ecole d'ingénieurs - Université de Toulon, Toulon, France
[3]Institute for Adriatic Crops and Karst Reclamation, Split, Croatia

**Correspondence:** Elena Terzić (eterzic@irb.hr)

**Abstract.**

In the Levantine basin, it has long been known that salinity can reach a maximum in a thin layer near the surface, particularly during the warm season when summer heating, evaporation, and low mixing prevail. This water mass has been linked to the generation of Levantine intermediate and deep waters, depending on winter heat loss and wind-induced mixing. However, a recent study demonstrated that similar conditions, referred to as 'surface saline lakes' (SSLs), can occur as far north as the Adriatic Sea. To investigate this, we analyzed data from Argo profiling floats across all Mediterranean basins, focusing on the upper layers (up to 200 m in depth), where such lakes are known to form. We developed an objective algorithm to detect SSLs within profiles, defining a SSL by a threshold-exceeding salinity gradient at its base. This definition allowed us to estimate SSL depth, SSL temperature and potential density anomaly ($PDA$) gradients at the base, and the Schmidt Stability Index which quantifies the energy needed to mix a SSL. We also ensured the quasi-continuity of Argo profiles throughout the year in our analyses, as SSLs are highly seasonal phenomena. SSLs exhibit minimum or vanishing occurrences between February and April, while peaking between August and October. SSLs were detected in all Mediterranean basins, with the highest prevalence—65–70% of profiles between July and December—occurring in the Levantine basin. During the August–October peak, SSLs exceeded 35% of monthly profiles in each basin, even in the Western Mediterranean, albeit with lower overall salinity levels and SSL variables ranges. These findings underscore the role of atmospheric heat and water exchange in all Mediterranean basins, influencing intermediate and deeper thermohaline properties through wintertime mixing. Despite pronounced interannual and seasonal variability, our analysis of data showed a significant positive trend in SSL depth, accompanied by decreasing thermohaline gradients (temperature, salinity, $PDA$) at SSL bases though the investigated period. The observed changes raise questions about their drivers—whether they indicate ongoing climate-change-induced salinization and shifts in Mediterranean water mass dynamics, or are merely manifestations of a multi-decadal variability.

## 1 Introduction

The Mediterranean Sea has been long time ago defined as a laboratory basin (Robinson and Golnaraghi, 1994) due to its specific bathymetry and a complex dynamics and processes that interact at different spatio-temporal scales, in particular in its eastern ultra oligotrophic part (Malanotte-Rizzoli et al., 2003). It is one of the climate change hot spots (Giorgi, 2006; Tuel and Eltahir, 2020), with increasing warming and salinification rates, especially in the Eastern Mediterranean (Borghini et al., 2014;



Kassis and Korres, 2020; Fedele et al., 2022; Aydogdu et al., 2023; Kubin et al., 2023). Sea surface warming rates have in the past 4 decades been 3-4 times higher compared to the global average (Pisano et al., 2020; Pastor et al., 2020), and climate projections predict a continuation of warming and salinization trends (Soto-Navarro et al., 2020), which could in turn affect the dense water formation efficiency (Somot et al., 2006; Parras-Berrocal et al., 2023; Verri et al., 2024).

The basin's thermohaline circulation follows a cyclonic pattern with numerous (sub-)mesoscale features (Millot and Taupier-Letage, 2005; Malanotte-Rizzoli et al., 2014). Less saline surface water flows from the Atlantic Ocean through the Gibraltar Strait due to the net freshwater deficit of the Mediterranean Sea, where evaporation surpasses precipitation, thus bringing surface salinities to extremely high values (up to and above 39). The Atlantic Water eastward flow in its journey undergoes a gradual increase in temperature and salinity. Upon reaching the easternmost part of the Mediterranean Sea, the Levantine

Surface Water, due to its extremely high surface salinity contributes to the formation of Cretan and Levantine intermediate waters (CIW, Velaoras et al., 2014; LIW, Malanotte-Rizzoli et al., 2003). Both of them are involved in the subsequent formation of deep water located in the Eastern Mediterranean, the Adriatic and Aegean Seas, but also reaching as far as the Western Mediterranean in the Gulf of Lions after crossing the Strait of Sicily (Millot, 2013; Margirier et al., 2020). Intermediate waters eventually form the bulk of the Mediterranean Outflow Water, therefore impacting the global ocean conveyor belt through

salinization of intermediate Atlantic layers (Aldama-Campino and Döös, 2020; Ayache et al., 2021).

Dense water formation events usually occur in late winter and early spring at the presence of cold and dry winds, such as bora in the Adriatic (Gačić et al., 2002; Mihanović et al., 2013) and mistral in the Gulf of Lions (Somot et al., 2018; Keller et al., 2024). At all dense water formation sites, the density of these waters is preconditioned by higher sub-surface and intermediate salinity values. Additionally, generation of intermediate and deep waters in the Levantine basin has been preconditioned by a

sharp maxima in salinity in the upper hundred or less meters, not documented in other basins, forming the so-called Levantine Surface Water (Kubin et al., 2019; Menna et al., 2022). However, a recent study demonstrated that similar conditions may occur as far north as the Adriatic Sea (Mihanović et al., 2021), with exceptionally high surface salinities driven by very low river discharges preconditioning the events for a year, higher-than-average heat fluxes during summer and early autumn periods, and higher-than-average evaporation minus precipitation rates, while the secondary salinity maximum persists in intermediate (up

to a few hundred of meters) layers coming from enhanced LIW and/or CIW horizontal flow.

Such changes might affect and modify the current thermohaline circulation patterns. For example, Li and Tanhua (2020) already documented a slow-down of deep ventilation in the Western Mediterranean, as well as a weakened Adriatic Sea source of ventilation in the Eastern Mediterranean. Moreover, Skliris et al. (2018) demonstrated a 20-30% increase in evaporation rates since the mid-20th century also due to lower precipitation and river discharges, further exacerbating sea surface salinities

and cases for changed dense water formation patterns and locations. LIW and CIV have also undergone both warming and salinification in the past decades that are faster than global rates (Schroeder et al., 2017; Potiris et al., 2024), at its formation source as well (Velaoras et al., 2015), the latter being due also to the reduced inflow of Black Sea freshened waters (Mamoutos et al., 2024). Furthermore, climate projections predict an increased stratification globally (IPCC, 2022; Holt et al., 2022; Roch et al., 2023) and in the Mediterranean (Llasses et al., 2018), which might further increase surface salinities and make vertical

mixing less efficient. All these factors combined may result in an increase in salinity profiles that have higher surface and



subsurface values due to a more stable water column and enhanced evaporation rates followed by increased warming, which could in turn modify the dense water formation paradigm, favoring double diffusive processes over convective mixing. Such vertical profiles with exceptionally high surface salinities, hereafter referred as surface saline lakes (SSLs), might be therefore an adequate indicator for the dense water preconditioning. In this work we attempted to characterize such vertical profiles by

introducing an objective algorithm to detect SSLs with the use of Argo float data set spanning for more than two decades. The international Argo Program has until 2019 already collected, processed, and distributed globally over two million vertical profiles of temperature and salinity from the upper 2000 meters (Wong et al., 2020). It is therefore a powerful tool in order to obtain comprehensive in-situ observations in locations that are also not so easily accessible. The Mediterranean Sea Argo network (Poulain et al., 2007) has collected data since 2001 with gaps only in the northern and middle Adriatic Sea due to its

shallow bathymetry and the Tunisian coast with the Gulf of Gabes (Fig.1). Such a high number of easily accessible vertical profiles in the entire basin enabled to study the spatio-temporal distribution of SSLs for the very first time. We introduce a robust algorithm for their detection and also assess possible changes in the past two decades, highlighting their causes and discussing if the observed trends are due to decadal variabilities or changes induced by a changing climate.

## 2 Methods

### 2.1 Argo Float Data

For this analysis, Argo profiling floats were downloaded for the entire Mediterranean Sea since their first deployments more than two decades ago (Fig. 1a). Only solid quality control (QC) profiles were chosen, i.e., QC=1 ("good") and QC=2 ("probably good") in delayed, but also real-time mode in order to include profiles from still active floats. A few additional conditions were set with the intention of discarding measurements that were not adequate for the analysis, despite the good QC flag. For

example, the first measurement needed to be at a depth shallower or equal to 10 m, as the analysis is primarily focused on surface layers. Moreover, the first depth quota had to be a positive number and all profiles needed to have measurements reaching at least 200 m. Only acquisitions in ascending mode were taken into consideration, and for the purpose of successful vertical gradients calculation (see next section), an additional condition was set that profiles should have only increasing depth values without any vertical oscillations. Variables of temperature ($T$ [$°C$]) and practical salinity ($S$ [-]) were used,

from which the potential density anomaly ($PDA$ [$kg\,m^{-3}$]) values were obtained with the use of TEOS-10 standards (https://www.teos-10.org).

### 2.2 Surface Saline Lake Detection Metrics

Surface saline lakes were calculated by setting a few conditions for each profile. SSLs' vertical salinity gradient $SG$ was flagged when it reached a threshold value lower or equal than $SG^{thres}$=-0.01 $m^{-1}$ within the first 200 m. The maximum depth

reaching or surpassing the threshold was then defined as the depth of $SG$, $zSG$. Concomitantly, values of temperature ($TG$) and density ($DG$) gradients were also stored. Additionally, the Schmidt Stability Index ($SSI$) was obtained as a proxy for the





water column stratification based on the approach in Duka et al. (2021), with the sole difference that the surface areas were not taken into consideration since we dealt with profiles and not surface areas. Saline lakes needed to meet a few further conditions. For example, salinity at the base of the saline lake had to be lower than the salinity value at the depth closest to the surface for at least 0.05, to avoid SSLs with low surface salinity coming, e.g., from river plumes. In order to reach statistically representative data in temporal terms, certain criteria needed to be met. Firstly, the number of profiles in a year must have been >1% of the total number of profiles in the whole period per each of the selected regions shown in Fig. 1a. Secondly, the number of profiles in each month of every year needed to be greater than 2% of the total profiles in that year. The annual number of profiles per each of the selected regions is shown in Fig. 1b.

## 3 Results

### 3.1 Upper Layer Climatology

Spatially, the collected Argo data was classified into 5 Mediterranean sub-regions, as shown in Fig. 1a: the Western Mediterranean, the Adriatic, Ionian, Aegean seas and the Levantine basin. Climatological median profiles of the upper 200 m of each of the locations, along with their 10-th and 90-th percentiles, can be seen in Fig. 2. Median temperature profiles range between 18-20$°C$ at the surface and around 14-16$°C$ at 200 m, where the lowest values are found in the Adriatic Sea, while the highest in the Levantine basin. The 90-th percentile is reaching up to 26$°C$ and 27$°C$ at the surface in all basins and 14.5 and 17$°C$ at 200 m in the Western Mediterranean and Levantine basin, respectively. The shallowest thermocline is present in the Adriatic Sea, presumably as the majority of floats are captured in the quasi-permanent cyclonic Southern Adriatic Gyre where upwelling is also persistent (Shabrang et al., 2016), while a subsurface temperature maximum can be detected in the Levantine basin. The median salinity profile exhibits highest values at the surface 50 m in the Levantine basin, with values above 39.0 and reaching more than 39.2 at the very surface (90-th percentile above 39.5), thus even in all-data averages showing a typical saline lake vertical structure. The Aegean Sea displays vertically a quasi-constant median value at around 39.1. This similarly holds also for the Adriatic Sea, although having lower median salinity values, between 38.8 and 38.9, and exhibiting a weak (sub)surface salinity minimum. The Ionian Sea and Western Mediterranean show a different vertical pattern, with salinity values increasing with depth, exhibiting the stronger influence of the Atlantic Water at the surface (Fedele et al., 2022). Median values for the Ionian Sea span between 38.7 at the surface and 39.0 at the bottom, and between 37.8 and 38.5 in the Western Mediterranean. Maximum surface median $PDA$ values are seen in the Adriatic Sea and Levantine basins, respectively, i.e., 28.1 $kg\,m^{-3}$ and 27.9 $kg\,m^{-3}$, with values reaching 29.0 $kg\,m^{-3}$ at 200 m for all basins. The 90-th percentile at the surface spans between 28.6 $kg\,m^{-3}$ and 29.1 $kg\,m^{-3}$ for the Western Mediterranean and Adriatic Sea, respectively.

As an example of the temporal evolution of thermohaline properties in the Levantine basin, a Hovmöller diagram for the long-term active WMO=6903269 float is shown in Fig. 3, along with its spatial trajectory color-coded in time (between September 2019 and October 2024). Surface salinities are for the first three years seen to have a seasonal variability with increasing values during early summer lasting until winter, when surface values are lower than at intermediate depths. Appearance of a surface salinity maximum is always in late spring, lagging for a month or two after the generation of strong seasonal thermocline. The





depths of both thermocline and halocline progressively deepened from ca. 10-20 m at their appearance to a hundred meters before exhibiting some of vertical mixing in winter, while salinity may surpass 39.3 at the surface. Below the halocline, waters with lower salinity (around or below 39.0) can be found, indicating remnants of Atlantic Water that may be tracked even to these farthest sections of the Levantine basin (Fach et al., 2021; Zodiatis et al., 2023). Further below, secondary salinity maximum can be found, fluctuating between 100 and 300 m, indicating LIW generated in previous winters and spreading horizontally over the basin (Kubin et al., 2019). Since mid-2022, however, there is a persistent SSL-like feature without any significant convective mixing or water column erosion that would bring surface values down or lower than at intermediate depths. Further, the subsurface salinity minimum is quite weak or nonexistent. This is especially seen in late winter and early spring 2024, where temperature, salinity and $PDA$ are more-or-less constant during the vertical mixing phase up to 300 m depth, i.e. $T$=19 °C, $S$=39.3 and $PDA$=28.3 $kg\,m^{-3}$. Indeed, the 2024 vertical mixing was stronger and reached deeper layers than during previous years.

### 3.2 Case Studies of Surface Saline Lakes

Examples of $T$, $S$ and $PDA$ profiles with their vertical gradients for different SSL cases is shown in Fig. 4. Autumn profiles in the Levantine basin (WMO 6901897, sampling date 28.11.2014) and Western Mediterranean (WMO 6903820, sampling date 15.10.2024), Fig. 4a and Fig. 4b, show deeper and more pronounced vertical salinity gradients. $SG$ values reach less than -0.04 $m^{-1}$ in both cases, with $TG$ values of approx. -0.3 and -0.6 °$C\,m^{-1}$ and $DG$ values of 0.04 and 0.13 $kg\,m^{-4}$ at the $zSG$ depths of 75.0 m and 39.6 m, respectively. Both examples are shown for a period after months of extensive evaporation and water column stabilization during summer. However, the major difference between these two profiles is in overall salinity values, i.e., the Levantine basin SSL has an approximately 0.8 higher salinity than the Western Mediterranean one. The lower limits of saline lake detection are seen in Fig. 4c for the Ionian Sea example (WMO 6990629, sampling date 02.06.2024) in late spring, when the lakes can start forming during the water column stabilization due to the existence of a strong thermocline and increased evaporation ($SG$ value of -0.011 $m^{-1}$, barely reaching the threshold value, at a depth of 19.8 m). The Adriatic Sea example shown in Fig. 4d, exhibits a shallow, but strong gradient in late summer (WMO 6903799, sampling date 17.08.2023), reaching $SG$, $TG$ and $DG$ values up to -0.04 $m^{-1}$ less than -0.8 °$C\,m^{-1}$ and above 0.2 $kg\,m^{-4}$, respectively, at a depth shallower than 15 m, with surface values surpassing 39.2, in accordance with findings from Mihanović et al. (2021).

### 3.3 Surface Saline Lakes Climatology

A spatial gradient can be seen in the distribution of SSLs, increasing in percentage towards east (Fig. 5). In the Eastern Mediterranean (the Levantine and Aegean seas), SSLs are present throughout the entire year, with a highest percentage in the Levantine (up to 72% in October). On the other hand, SSLs are reduced in numbers after convective mixing during winter/early spring or completely disappear in the Western Mediterranean, where the highest percentage does not reach a number higher than 35% in September. Overall, a highest percentage is seen in late summer and early autumn months after having reached a longer period of stable stratification and enhanced evaporation, peaking in September/October for most basins. The percentage





of SSLs is in general highest in the easternmost region - the Levantine basin, above 40% from June to December, followed by the Adriatic, Ionian and Aegean seas and the Western Mediterranean.

Monthly climatology of surface saline lakes depth, $zSG$, displays lowest values in the Adriatic Sea and the Western Mediter-
ranean, whilst deepest gradient locations are found in the Levantine basin and Aegean Sea (Fig. 6a). Both $TG$ and $DG$ exhibit strong seasonal variability, clearly showcasing the temperature's influence in determining density in the surface layers. Minimum (maximum) values of $TG$ ($DG$) can be seen in the Adriatic Sea (between July and September, Figs. 6b, d), denoting shallow but intense accumulation of heat near the surface. Monthly median $TG$ values span between close to zero and -0.04 $^\circ C\ m^{-1}$ during summer, reaching peaks in the Adriatic below -0.6 , even -0.8 $^\circ C\ m^{-1}$, whereas $DG$ are up to 0.1 and 0.2 $kg\,m^{-4}$ in the Adriatic. $SSI$ shows greatest values during winter months, similar as in $zSG$, i.e., up to 1200 $J\,m^{-2}$ (Fig. 6e). This might be due to the index's sensitivity to low surface $PDA$ values in the Western Mediterranean which could be caused by an enhanced influence of fresher Atlantic Water. $SG$, on the other hand, does not show any seasonal variability, with median values ranging between -0.018 and -0.013 $m^{-1}$, except the Adriatic Sea value on January (Fig. 6d), but this value is based on a limited number of SSLs. High $SSI$ during winter in the Levantine coincides with greatest $zSG$, therefore showing that certain profiles are not fully eroded during winter cooling and convection, but just stretch SSL towards greater depths (Fig. 6e). The overall median ranges for individual basins are: from -0.21 $^\circ C\ m^{-1}$ (Levantine) to -0.46 $^\circ C\ m^{-1}$ (Adriatic) for $TG$, between 0.04 (Levantine) to 0.11 (Adriatic) $kg\,m^{-4}$ for $DG$ and between 31.6 (Adriatic) to 100 $J\,m^{-2}$ (Ionian) for $SSI$.

Probability density functions (PDFs) for each of the SSL detection and description parameters per region can be seen in Fig. 7. $zSG$ peaks at 20 m in the Adriatic Sea, 35 m in the Western Mediterranean, while around 45 m in other basins. The widest spread is in the Aegean Sea, with high PDF values to depths of approx. 75 m. This might indicate a difference in SSLs between northern and southern Aegean Sea, as the profiles are quite rare in the separation area in the middle Aegean Sea populated with a number of islands (Fig. 1a). Unlike for $zSG$, $TG$ displays a widest spread of PDF in the Adriatic Sea (Fig. 7) and its shape differs substantially compared to other basins, showing a more evenly distributed range of values. In all basins the peak is at slightly lower or around -0.1 $^\circ C\ m^{-1}$. In terms of $SG$, the PDFs do not differ much from basin to basin, all of them peaking at near -0.015 $m^{-1}$ and having a relatively narrow distribution range. As seen in Fig. 6, $DG$ is more influenced by $TG$, hence the PDF is also more variable between regions, with the Adriatic Sea again having a much wider spread and a flatter peak. This is followed by the Western Mediterranean, whereas the Levantine basin, Ionian and Aegean seas all peak around 0.02 $kg\,m^{-4}$ and have a very similar distribution shape. $SSI$ shows a flattest distribution range for the Aegean Sea, followed by the Western Mediterranean. The Adriatic and Ionian seas display most similar curves, with the Ionian region peaking at higher $SSI$ values compared to the former. The narrowest PDF is seen for the Levantine basin.

## 3.4 Interannual variability and trends

Monthly median time series exhibit a high variability between basins (Fig. 8). The Adriatic Sea displays overall lowest (highest) $TG$ ($DG$) and $zSG$ values, most notably so since 2015, with a continuous decrease (increase) of $TG$ ($DG$) with time up to present. The Levantine basin, the Aegean and Adriatic seas show a deepening of $zSG$ since 2015, highlighting a drastic jump between 2014 and 2015 (maximum values increasing from 90 to 175 m for the Levantine Sea). This shift is concomitant with





an increase in $SSI$, also peaking between 2015 and 2021, after which the values are returning back to the previous or a bit higher range of values up until present.

Due to temporal gaps when looking at each region separately, trends were obtained for the whole Mediterranean Sea. The seasonal signal was removed with annual and semi-annual cosine function fit. Results reveal a deepening of SSLs in time, with $zSG$ increasing around 1 m/year. Vertical gradients of thermohaline properties all exhibit a decreasing trend: $SG$ around 0.0002 $m^{-1}$ per year, $TG$ = 0.0045 $°C\ m^{-1}$ per year and $DG$ = -0.0013 $kgm^{-4}$ per year (Fig. 9). $SSI$ has a statistically insignificant increasing trend (p>0.01) of 4.81 $Jm^{-2}$ per year, with an interannual dynamic like the one documented for $zSG$.

## 4  Discussion

For the first time, an objective analysis of in situ ocean data indicates that the accumulation of salt near the sea surface during strong heating, evaporation and low mixing seasons is present in the whole Mediterranean Sea. Inside the Mediterranean itself, such salinity profiles have been insofar substantially seen only in the Levantine basin (Manca et al., 2004; Ozer et al., 2020; Kubin et al., 2019), while some case studies document their occasional existence in the Adriatic Sea (Mihanović et al., 2021). The accumulation shows some common characteristics (e.g., halocline strength) and some differences (e.g., the overall temperature and salinity) between different Mediterranean basins, but SSLs are less frequent in all regions compared to the Levantine basin. This implies that now there exist other regions inside the Mediterranean where surface salt gets transported in the intermediate depths due to similar process as in the Levantine area (through wintertime mixing or double diffusivity), with an albeit weaker dynamics, such as has been already documented in the Adriatic Sea (Amorim et al., 2024).

Results have shown a great deal of seasonal and spatial variability of surface salt accumulation, but also interannual variations that resulted in small, but statistically significant trends. A west-to-east spatial gradient in both depth and frequency of SSLs may be explained with the differences in atmospheric patterns, as well as in general surface salinity values. The Western Mediterranean is less saline due to its proximity to the Atlantic and therefore stronger effects of less saline Atlantic waters, as well as a higher precipitation rate, higher riverine discharge and lower evaporation. The Eastern Mediterranean (except the Adriatic Sea) is more arid with less freshwater output, getting some freshwater input through the Nile delta, but is generally a region with much less precipitation, higher evaporation and less wintertime mixing (Ulbrich et al., 2012), all contributing to higher surface salinities.

In terms of interannual variability, the drastic increase in certain properties that define the SSLs (such as $zSG$ and $SSI$) since 2015 could highlight a change in the atmospheric forcings in the last years. However, results could be biased due to spatio-temporal variable float sampling, as well as by the varying annual number of floats. As shown in Fig. 1b, the annual number of floats drastically increased from 2013 onward, but the "jump" seen in Figs. 8a, d and e appeared only from 2015. The explanation that results might not be biased due to varying sampling rates and profile numbers could be further argued when looking at the case of the Ionian Sea (light brown line in Fig. 1b), where the high increase in the number of profiles in 2024 does not result in an increase in $zSG$ or $SSI$ as in 2015 in the Levantine and Adriatic Seas, and a similar increase in the Aegean Sea following in 2016. The Western Mediterranean does not have any such changes, hence it's a question whether





this is the result of localized changes in the eastern Mediterranean basins, where trends in both warming and salinification
have been rapidly increasing in the past decades (Aydogdu et al., 2023). For example, this has been seen also in the Cretan
Sea, especially since 2017 (Chiggiato et al., 2023). During the last salinification phase since 2017, the high salinity increase
extended down to depths below the intermediate layer, and it is not yet clear which implications it might have had or will have
in the following years. Consequences have been felt in the Southern Adriatic too, facilitating diffusivity of more saline waters
originating from the Levantine basin towards the bottom of the pit (Mihanović et al., 2021; Amorim et al., 2024).

In the regional climate projections, net heat loss is expected to decrease (Soto-Navarro et al., 2020) due to an increase in
shortwave, net-long wave and sensible heat loss (Dubois et al., 2012). This could have already started resulting in a weakened
convection that was unable to efficiently mix the water column, as seen in the Hovmöller diagram in Fig. 3 in the Levantine
basin. Enhanced evaporation and reduced precipitation may both contribute to higher surface salinities, thus requiring more
energy for an efficient vertical mixing. This has been already demonstrated as a trend in the Mediterranean Sea (Skliris et al.,
2018), and it's expected to continue also in the future (Soto-Navarro et al., 2020). In combination with reduced convective
mixing, such features could persist over the entire year without being eroded from one winter season to the next, as it's already
the case in subtropical gyres in the Atlantic and Pacific Oceans and will be further explained below.

SSLs might therefore serve as an additional indicator of climate change, given the fact that an increased stratification (and
hence less intense mixing) is to be expected in the future as one of the consequences of continuous warming. According to the
Special Report on the Ocean and Cryosphere in a Changing Climate (IPCC, 2022), the global stratification has increased in
1998-2019 compared to 1979-1990 for around 2.3% and might increase up to 12-30% compared to the 1985-2005 reference
period at the end of the 21st century (2081-2100) in case of a business-as-usual RCP8.5 scenario. Globally, Cheng et al.
(2020) estimated an accelerating increase in average salinity since 1960s for the first 2000 meters (2-4 % per $°C$ of warming).
Salinity variations can thus sensitively reflect the net exchange of freshwater between the ocean and the atmosphere. Similar
mechanisms were observed in the Adriatic Sea, as demonstrated in Mihanović et al. (2021) - after the thermocline formation,
the halocline appears after a few months due to the evaporation of a shallow surface layer. The stronger the warming rate, the
greater the thermocline and hence higher evaporation rate, which as a consequence causes higher surface salinities.

The Mediterranean Sea is quite specific due to its exceptionally high surface salinities. In case of a global salinification
trend, changes in the region may potentially provide a clue about the possible changed thermohaline dynamics in other parts of
the world and could thus justify its being denoted as a laboratory region. The question remains whether the currently periodic
SSLs may become a (quasi-)permanent feature in the absence of strong winds that could erode the vertical profile.

Outside the Mediterranean Sea, surface saline lakes may be (quasi-)permanent features that are present all year round
also in the subtropical high atmospheric pressure zones with high evaporation, low precipitation and low vertical mixing
induced by winds. Such cases are present both in the Atlantic and Pacific oceans. For example, two float trajectories in the
Northern Pacific (WMO=4903503, https://fleetmonitoring.euro-argo.eu/float/4903503) and Southern Pacific Subtropical Gyres
(WMO=5905261 https://fleetmonitoring.euro-argo.eu/float/5905261), regions of North and South Pacific High pressure sys-
tems, both exhibit the SSLs as a permanent feature with stable layers of higher (sub-)surface salinities reaching less than 100 m
in the Southern and more than 200 m in the Northern Gyre, albeit maximum salinities do not surpass 35.3 in the North and 36.4



in the South. The Atlantic is generally influenced by the intrusion of highly saline waters through the Mediterranean Outflow,
which causes it to be saltier compared to the Pacific. A float in the Northern Atlantic Subtropical Gyre, a region of the Azores
High pressure system (WMO=4903739, https://fleetmonitoring.euro-argo.eu/float/4903739, upon traveling southward, the sur-
face salinity increases to values up to 36.2, exhibiting a permanent and stable layer of higher subsurface salinities. Similarly, in
the Southern Atlantic, the region of the St. Helena High (WMO=6904187, https://fleetmonitoring.euro-argo.eu/float/6904187),
surface salinity reaches up to 36.6, with a SSL shape stable throughout the entire sampling period. Clearly, no seasonality is
seen in such regions, and SSLs persist as a stable, constant feature without being eroded. That is the major difference to the
current climate of SSLs in the Mediterranean, which still erode in majority of the basin during wintertime mixing and therefore
contribute to the generation of intermediate and deep waters and their ventilation.

This method could be in principle extended to other regions, however it should be noted that the salinities in question in the
Mediterranean Sea surpass the ones in the other regions by up to 3 or even 4. Maximum salinities in the Pacific cases were
between 35.3 and 36.4, whereas values in the Eastern Mediterranean, as well as exceptionally in the Adriatic Sea, may reach
even 39.4-39.5, as shown in Figs. 2 and 3. This is an important difference, showing that even though SSLs might be present
in other places, their sheer existence is not a sufficient condition for a potential sinking and dense water formation, as in the
Mediterranean. Still, this might eventually change (at a certain level, presumably not like in the Mediterranean) in oceans in
the future climate, as subtropical high systems are also projected to change in space, time and intensity (Cherchi et al., 2018).

Linear trends from a time series built from the Mediterranean Argo data after the seasonal signal removal exhibit weakening
vertical gradients of temperature, salinity and $PDA$, while at the same time showing a deepening of the SSLs (increasing
$zSG$). This could also be an indicator that the water mass under the SSL is getting saltier and warmer, thus weakening the
gradients and enabling an easier expansion of SSLs. One of such examples has been already documented and described in
Schroeder et al. (2017), where LIW was shown to get both warmer and saltier at the Sicily Channel at intermediate depths.

Finally, SSLs could for all these reasons be a possible indicator of changing biogeochemical properties and therefore may
affect marine life. The first strong appearance of SSLs in the Adriatic in 2017 (Beg Paklar et al., 2020) substantially changed
microbial food web, e.g. detouring the abundance of heterotrophic bacteria for several standard deviations from the average
values, and changing the composition rates between Prochlorococcus, Synechococcus and picoeukaryotes. Salinization of
the Eastern Mediterranean and Southern Adriatic already resulted in a feedback mechanism which completely switched the
oxygen content dynamics in the northern Ionian Sea (Martellucci et al., 2024) between different phases of the Adriatic-Ionian
Bimodal Oscillating System known - at least up to present - to drive Adriatic thermohaline and biogeochemical properties
(Gačić et al., 2010; Civitarese et al., 2010, 2023; Batistić et al., 2014). In the Mediterranean and globally, projected changes in
stratification would decrease dissolved oxygen and primary production (Doney et al., 2014; Richon et al., 2019; Lachkar et al.,
2024), undoubtedly having strong effects propagating the signal through the food web. Surface saline lakes might be one of
mechanisms fostering such changes.



# 5   Conclusions

In this work we introduced a robust and objective algorithm for the detection of upper-layer profiles in different Mediterranean basins characterized by a quasi-constant maximum in salinity above the pycnocline, referred as surface saline lakes (SSLs). This phenomenon has been known to occur regularly in the Levantine basin and sporadically documented for some others,

like the Adriatic Sea. First, we showed different examples of vertical profiles where SSLs were spotted, ranging in gradient strength and depth. The climatological analysis showed highest SSLs percentage and their maximum depth in the Eastern Mediterranean, especially the Levantine basin. Minimum occurrences were seen between February and April, while peaking between August and October.

Lakes were present throughout the entire year in the Eastern Mediterranean, whereas in the Western Mediterranean they

disappeared in late autumn or early winter, most likely due to the convective erosion of the water column. These findings underscore the role of atmospheric heat and water exchange in all Mediterranean basins, influencing deeper thermohaline properties through winter mixing.

SSLs were detected in all Mediterranean basins, with the highest prevalence occurring in the Levantine region, i.e., 65–70% of profiles between July and December. During the August–October peak, SSLs exceeded 35% of monthly profiles in each

basin, even in the Western Mediterranean, albeit with varying overall salinity levels and SSL variables ranges. By having looked at the monthly climatology of various SSL definition parameters, a small range of variability was observed in salinity gradients at the SSL base, unlike for temperature and density gradients, the latter seemingly influenced more by temperature than salinity. SSL depth monthly values ranged between 25 m during spring and summer months to up to 175 m in late winter, concomitantly with high $SSI$ values, showing that certain SSLs are deep enough that are not eroded by winter convection.

Despite pronounced interannual and seasonal variability, our analysis of data showed a significant trend in SSL depth, accompanied by decreasing thermohaline gradients (temperature, salinity, $PDA$) at SSL bases though the investigated period. However, these trends may partly reflect sampling biases due to time-space differences in Argo float coverage, which has been substantial before 2013. The observed changes prompt questions about their underlying causes, as the observation period is still too short to derive any robust conclusions—whether they reflect ongoing climate-change-driven salinization and alterations in

Mediterranean water mass dynamics or simply result from natural decadal variability. In any case, the SSLs may be an indicator of changes within the upper ocean where stratification is globally projected to occur, undoubtedly with substantial effects to biogeochemistry and marine life.

*Code availability.*   Codes are freely available upon request.

*Data availability.*   All data was downloaded through Argopy - a Python library aimed at Argo data users: https://pypi.org/project/argopy/



*Author contributions.* ET: Formal analysis, Methodology, Investigation, Data Curation, Writing - Original Draft, Review and Editing, CG: Formal analysis, Investigation, Data Curation, Writing - Review and Editing, IV: Conceptualization, Methodology, Investigation, Writing - Original Draft, Review and Editing, Supervision, Funding Acquisition

*Competing interests.* Authors declare that there are no competing interests.

*Acknowledgements.* To Euro-Argo data providers developers and their great work for making the used products and the data available
for research. We'd also like to thank ArgoPy library developers for making the Argo data analysis much easier and user-friendlier. The research has been supported by Croatian Science Foundation through project GLOMETS (Grant IP-2022-10-3064) and incoming mobility scheme for postdoctoral researchers (MOBDOL-2023-12, Elena Terzić), as well as by the Interreg Italy-Croatia project AdriaClimPlus (Grant ITHR0200333).



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





**Figure 1.** (a) The Mediterranean Sea geography, with color-coded time span of Argo profiles used in the research and marked regions over which the analyses have been carried out: the Adriatic Sea, the Ionian Sea, the Aegean Sea, the Levantine basin and the Western Mediterranean (image background from ©Google Maps). (b) The annual number of of Argo profiles per basin (b).





**Figure 2.** Median values with the 10-th and 90-th percentiles for T, S, PDA (columns) per basin (rows).



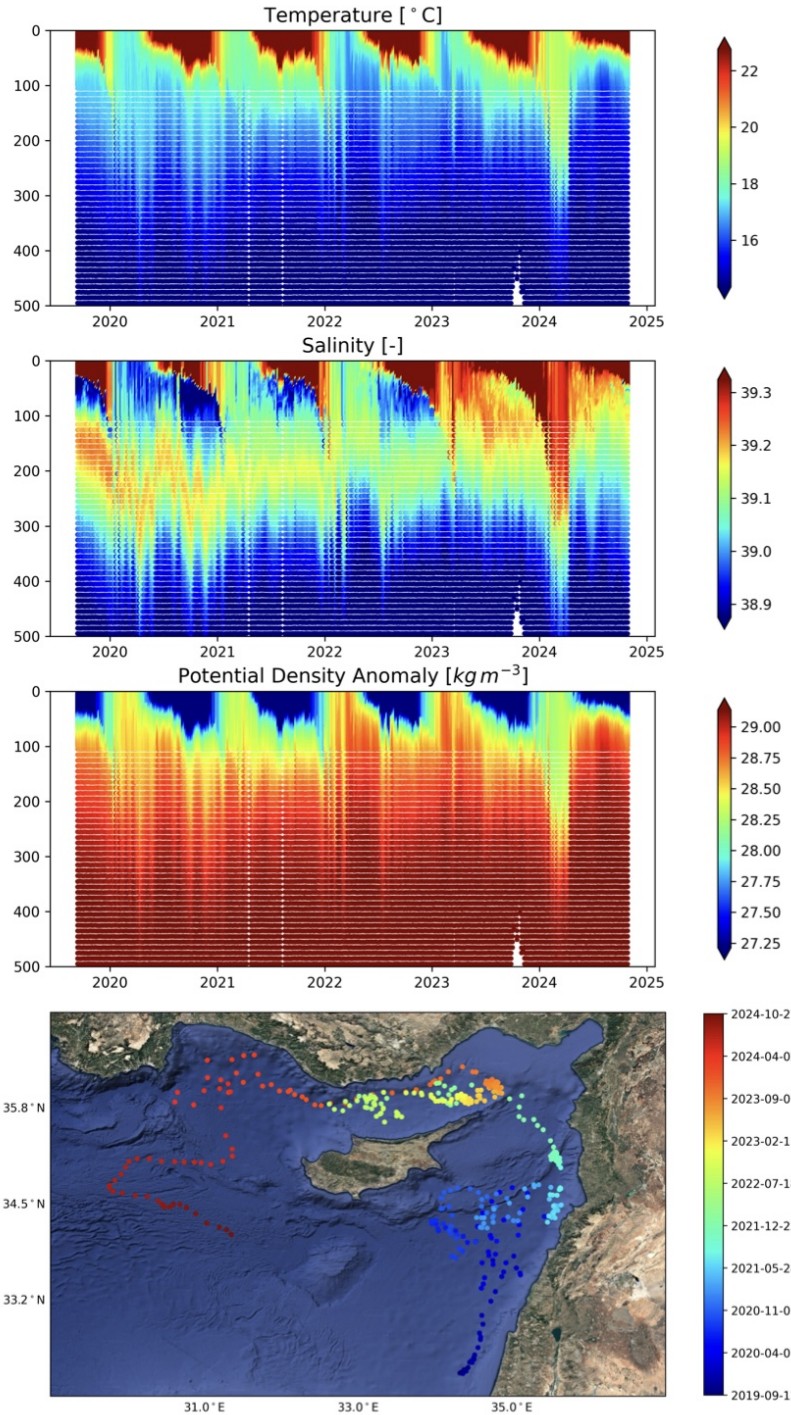

**Figure 3.** Hovmöller diagram of T, S and PDA for WMO=6903269 in the Levantine basin, with its trajectory shown at the bottom (image background from ©Google Maps).





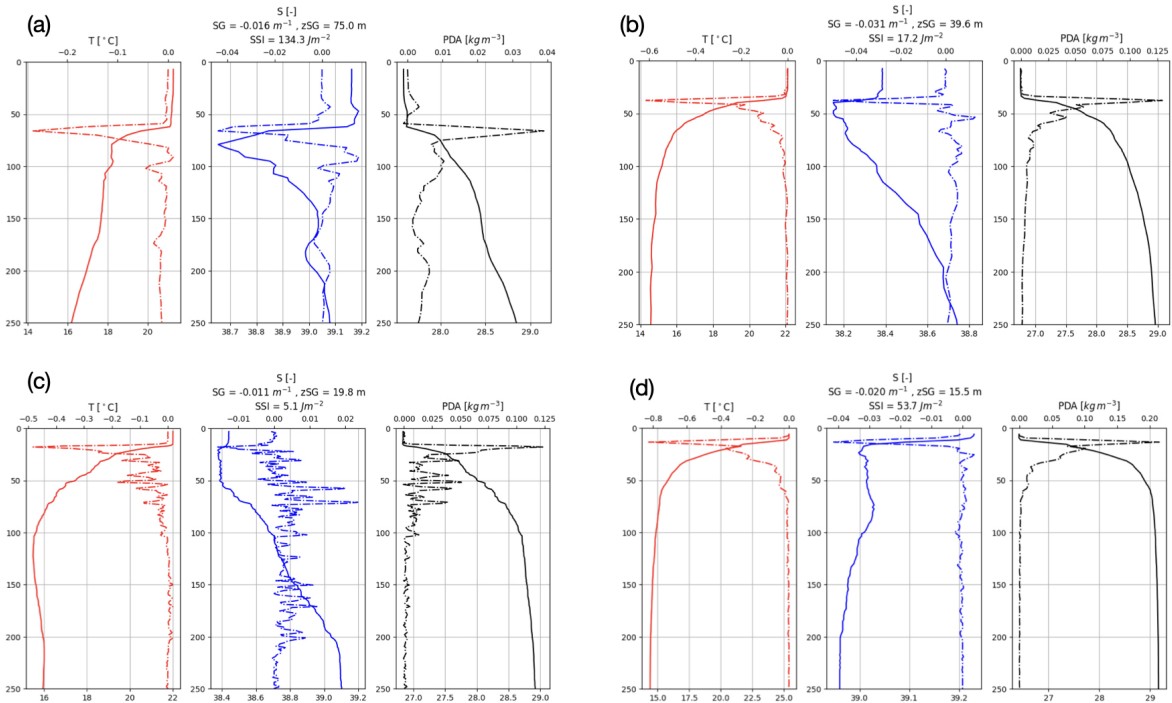

**Figure 4.** Examples of vertical T, S and PDA profiles (full lines) and their vertical rates of change (dashed lines) during which a surface saline lake is detected: a) the Levantine basin (float number 6901897, profile sampled on 28.11.2014), b) the Western Mediterranean (float number 6903820, profile sampled on 15.10.2024), c) the Ionian Sea (float number 6990629, sampled on 02.06.2024) and d) the Adriatic Sea (float number 6903799, sampled on 17.08.2023).

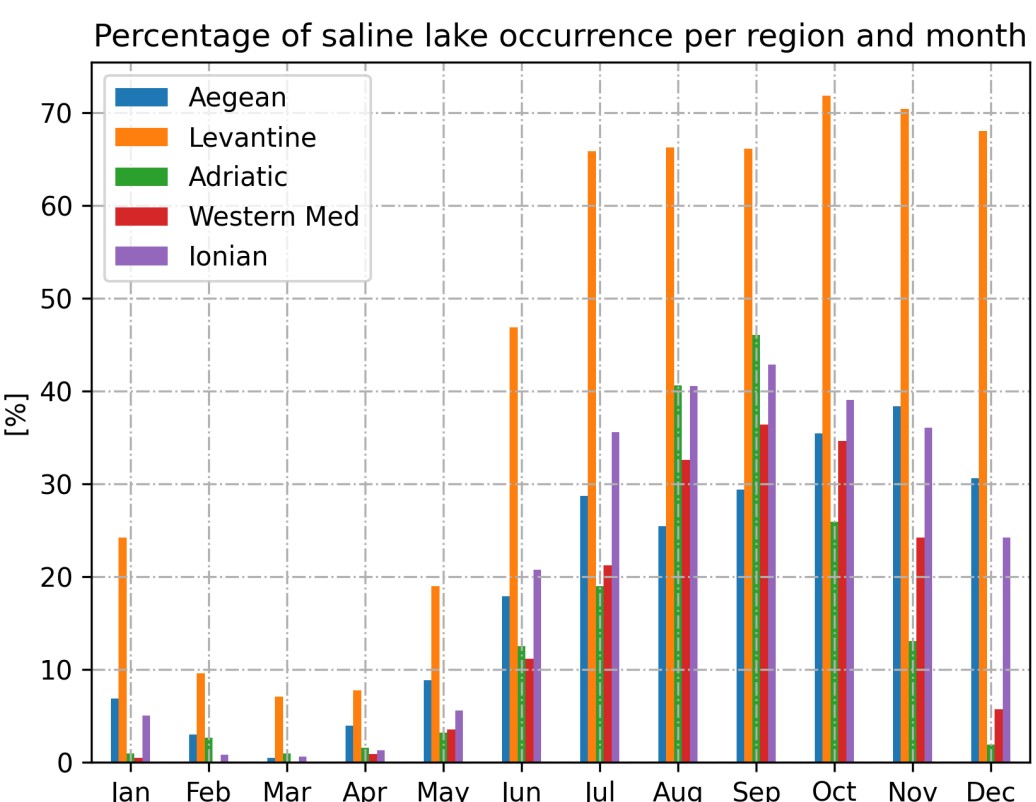

**Figure 5.** Percentages of profiles in which surface saline lakes occur for all months and basins.







**Figure 6.** Monthly averages in (a) the depth of surface saline lakes $zSG$, the respective (b) $TG$, (c) $SG$ and (d) $DG$ at the base of the lake, and (e) $SSI$ of a lake over the selected regions.



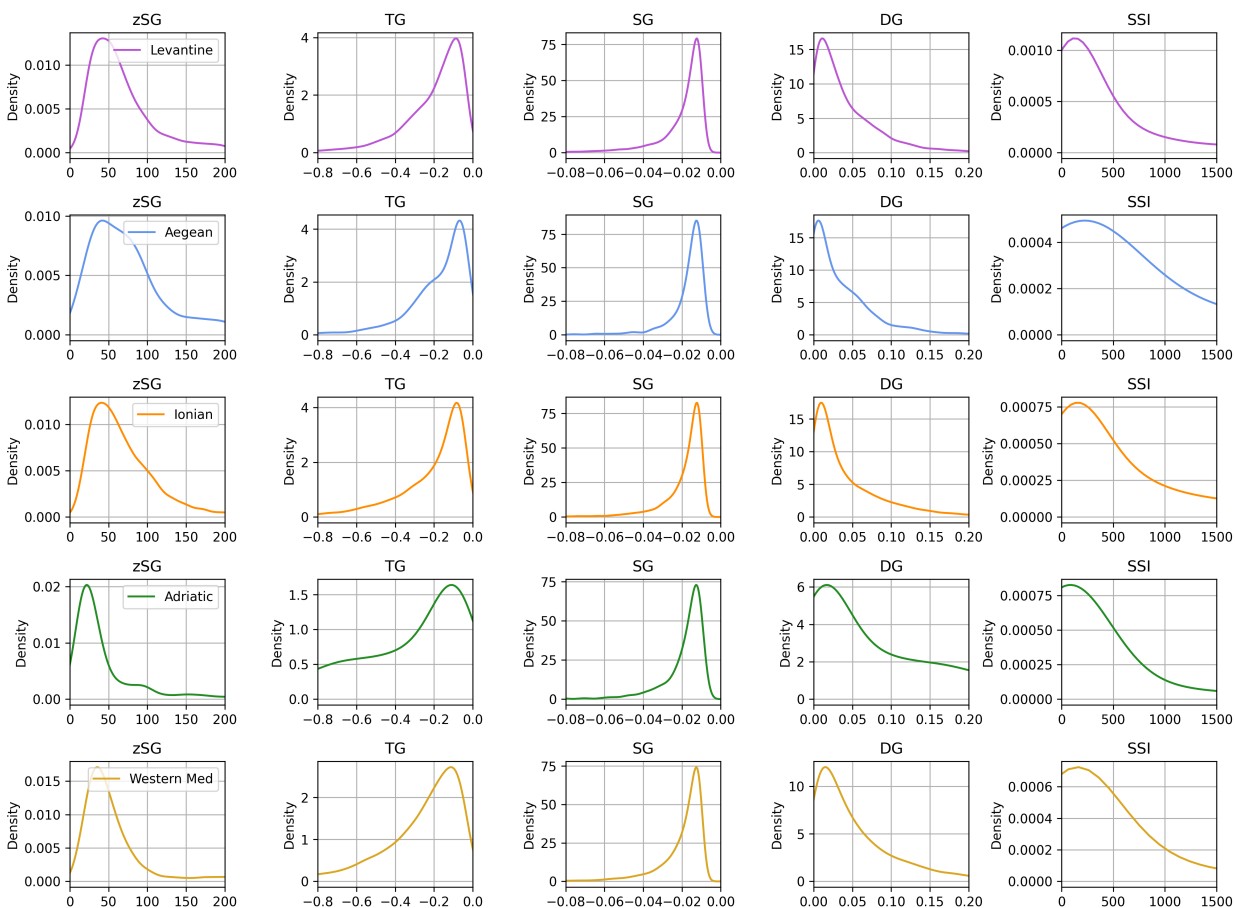

**Figure 7.** Probability density functions for: $zSG$, $TG$, $SG$, $DG$ and $SSI$ (columns) of a lake over the selected regions (rows).





**Figure 8.** Monthly median time series in (a) the depth of surface saline lakes $zSG$, the respective (b) $TG$, (c) $SG$ and (d) $DG$ at the base of the lake, and (e) $SSI$ of a lake over the selected regions.





**Figure 9.** Linear trends in (a) the depth of surface saline lakes $zSG$, the respective (b) $TG$, (c) $SG$ and (d) $DG$ at the base of the lake, and (e) $SSI$ of a lake.