# Peer review of "Surface saline lakes in the Mediterranean Sea"

_EGUsphere, 2025_

## Referee Comment (RC1)

In this study, Terzić and colleagues examine the spatial distribution and temporal variability of surface saline lakes (SSLs) across the Mediterranean Sea, ranging from seasonal changes to long-term trends. A detection metric based on Argo float data is employed to characterize SSLs, as well as associated temperature and density patterns. The results indicate that SSLs are present in nearly all Mediterranean sub-basins. Furthermore, a significant upward trend in SSL depth is observed over the study period. Longer-term observations are likely required to determine whether this trend reflects multi-decadal variability or ongoing salinization driven by climate change. The manuscript is well-structured and easy to follow overall. However, there are several areas that require improvement in terms of clarity and presentation. Some points are summarized as follows.

Based on these concerns, I recommend the manuscript for publication with major revisions.

Introduction
While the article investigates SSL characteristics, the introduction begins with general physical background and then shifts focus to thermohaline circulation. Although SSLs and deep-water formation are indeed intertwined, introducing SSLs (Line 63) in the final paragraph of the introduction may be inappropriate. A more SSL-focused introduction is recommended.

L70: Gulf of Gabes
Mark the geographic location in Fig. 1. Additional locations mentioned in the text could also be included in Fig. 1 or another figure to aid readers unfamiliar with the region.

L77-78: QC=1, QC=2
How reliable is Argo profiling data labeled with QC=2, and how is this category of data typically treated? What proportion of the total dataset falls under QC=2? Would limiting analysis to only QC=1 data significantly influence conclusions, such as those related to seasonal patterns or long-term trends?

L96-98: 1%-2%
These two criteria were applied to ensure a minimum level of data coverage for each month of each year. Is there any justification provided for selecting these specific thresholds, or has any sensitivity analysis been performed to test their impact?

L133-135: intensified mixing
More explanation of the intensified mixing and its penetration into deeper layers in late winter and early spring 2024 would be beneficial.
In addition, consider moving the WMO=6903269 float case (Hovmöller diagram) to Section 3.2, as the title of Section 3.1 focuses on climatology.

L159: "lowest values in the Adriatic Sea"
zSG values are indeed low during most months in the Adriatic Sea, but they peak in March. Are you referring to the annual mean of the monthly climatology? Please clarify.

L165-170: "SSI"
The author attributes the higher SSI during winter in the Western Mediterranean to low surface PDA values caused by fresher Atlantic Water, and in the Levantine and Aegean Seas (not mentioned) to deeper zSG. However, although an increase in zSG from September to December is evident, it is not reflected in the SSI.

L184-185:
The SSI peak in the Adriatic Sea appears visually higher than in the Ionian Sea. Additionally, the conclusion that the Adriatic and Ionian Seas exhibit the most similar curves seems somewhat arbitrary.

L195-197:
Since the significance of the linear trend depends on the data distribution, you should state your assumption or show the distribution.

L216-217:
The sharp rise in zSG is unlikely from sampling, as discussed by the author. Atmospheric forcing is mentioned but not well explained

L237: "…the case in subtropical gyres in the Atlantic and Pacific Oceans"
Any reference?

Consider increasing the font size in Figure 4 for better readability.

---

## Author Comment (AC1)

To Reviewer#1: thank you for all your instructive suggestions that made this version of the manuscript much better. Smaller remarks were all incorporated in the updated version of the manuscript (see changes tracked in **blue**), whereas detailed answers to specific questions are addressed below.

*Introduction*
*While the article investigates SSL characteristics, the introduction begins with general physical background and then shifts focus to thermohaline circulation. Although SSLs and deep-water formation are indeed intertwined, introducing SSLs (Line 63) in the final paragraph of the introduction may be inappropriate. A more SSL-focused introduction is recommended.*

We rewrote the Introduction by taking into consideration this recommendation, which was suggested also by Reviewer#2.

*L70: Gulf of Gabes*
*Mark the geographic location in Fig. 1. Additional locations mentioned in the text could also be included in Fig. 1 or another figure to aid readers unfamiliar with the region.*

We added the Gulf of Gabes and other geographic locations in the new Fig.1.

*L77-78: QC=1, QC=2*
*How reliable is Argo profiling data labeled with QC=2, and how is this category of data typically treated? What proportion of the total dataset falls under QC=2? Would limiting analysis to only QC=1 data significantly influence conclusions, such as those related to seasonal patterns or long-term trends?*

We checked the used data and out of 67.554 profiles in total, 64.809 profiles have a QC=1 (good) and only 2.745 are with QC=2 (probably good), which is around 4% of the entire data set.
Therefore, removing QC=2 data doesn't significantly impact results, any of the seasonal or long-term trends and conclusions, as the percentage is rather low. We put this information and further clarifications in the revised manuscript.

*L96-98: 1%-2%*
*These two criteria were applied to ensure a minimum level of data coverage for each month of each year. Is there any justification provided for selecting these specific thresholds, or has any sensitivity analysis been performed to test their impact?*

This percentage was taken by a choice which is based on balancing the length of the time series and the availability of enough data for analyses in a particular year or month. For example, from 2002 to 2024, the time span is 23 years. Putting a threshold on 2% instead of 1% would have meant discarding a larger number of data from the beginning of the time series, given its temporal distribution which has much less data in the first few years (as shown in Fig.1 - bottom). However, if we reduce it to less than 1%, that would have affected the quality of our calculations, as resulting in a disballance in available data between different years. 1% was therefore taken as a fair compromise. Similarly seasonal computations - having 12 months every year, which is a half of the number of years (23), we doubled the threshold percentage from 1 to 2%. We put the appropriate clarification in the revised manuscript.

*L133-135: intensified mixing*
*More explanation of the intensified mixing and its penetration into deeper layers in late winter and early spring 2024 would be beneficial.*

Thank you for having raised this point, which is important to discuss when talking about SSLs trends.
We downloaded ERA5 hourly reanalysis data between 2002 and 2024 for the Levantine region (latitudes between 33 and 36 degrees and longitudes between 22 and 33 degrees).
We obtained winter (January-February-March) daily anomalies for 2024, subtracting the 2002-2024 baseline, for 4 components: latent, sensible, shortwave and longwave fluxes, as well as calculated their sum.
This resulted in negative total anomalies in January and beginning of February, implying surface cooling that in turn causes vertical convection to occur.
We added this figure and also an explanation in the text.

[Figure]

*In addition, consider moving the WMO=6903269 float case (Hovmöller diagram) to Section 3.2, as the title of Section 3.1 focuses on climatology.*

We moved the Hovmöller diagram and its comment to Section 3.2.

*L159: "lowest values in the Adriatic Sea"*
*zSG values are indeed low during most months in the Adriatic Sea, but they peak in March. Are you referring to the annual mean of the monthly climatology? Please clarify.*

We corrected this sentence, thank you for pointing this out. We meant overall, but overlooked the values in March. For this reason, we made a modification to the plot in order to enhance its clarity - grid lines were moved in such way that each group of points per month is between two vertical grid lines.

*L165-170: "SSI"*
*The author attributes the higher SSI during winter in the Western
Mediterranean to low surface PDA values caused by fresher Atlantic Water,
and in the Levantine and Aegean Seas (not mentioned) to deeper zSG.
However, although an increase in zSG from September to December is
evident, it is not reflected in the SSI.*

Indeed, SSI is dependent on both the density, as well as on the zSG range.

$$SSI = g \sum_{i=z_0}^{z_{SG}} (z_i - z_g)(\rho_z - \rho_i)\Delta z_i$$

Hence, even though an increase in zSG could be present, the density
decrease and changes within a surface saline lake could be the prevailing
mechanism for these particular months, which could be due to a higher
precipitation rate, riverine discharge, advection of water masses, as written
in the draft. We added an explanation in the updated version of the
manuscript.

*L184-185:*
*The SSI peak in the Adriatic Sea appears visually higher than in the Ionian
Sea. Additionally, the conclusion that the Adriatic and Ionian Seas exhibit the
most similar curves seems somewhat arbitrary.*

This is also a good point which was missed due to varying plot limits and
was mentioned also by Reviewer #2. We rephrased this sentence
accordingly and modified Figure 7, which now exhibits much clearer and
varying shapes.

*L195-197:*
*Since the significance of the linear trend depends on the data distribution,
you should state your assumption or show the distribution.*

When computing the linear regression, we used Python's
*scipy.stats.linregress* through which we obtained also the p-value using Wald
Test with t-distribution of the test statistics. The latter assesses constraints
on statistical parameters, so we consider it appropriate for yearly data
distributions of our surface saline lakes variables. In our analysis, p-values

were below 0.05 for all variables considered, but slightly above 0.01 for SSI, as already mentioned in the manuscript.

*L216-217:*
*The sharp rise in zSG is unlikely from sampling, as discussed by the author. Atmospheric forcing is mentioned but not well explained*

See a more detailed reply above for L133-135. We added more details regarding the atmospheric forcing in the updated version of the manuscript.

*L237: "…the case in subtropical gyres in the Atlantic and Pacific Oceans"*
*Any reference?*

We added a reference.

*Consider increasing the font size in Figure 4 for better readability.*

We followed this suggestion and enlarged the font size, thank you.

---

## Author Comment (AC2)

To Reviewer#2: thank you for all your instructive suggestions that made this version of the manuscript much better. Smaller remarks were all incorporated in the updated version of the manuscript (see changes tracked in **green**), whereas detailed answers to specific questions are addressed below.

*Review of the manuscript "Surface saline lakes in the Mediterranean Sea", by Elena Terzić et al.*

*In this study the authors analyze the distribution and variability of Surface Saline Lakes (SSLs) in the Mediterranean Sea, using observational data from Argo float profiles. The authors use an algorithm to detect and characterize SSLs across the basin, which allow them to characterize the seasonality, interannual variability and long-term trends of these events over several Mediterranean sub-basins. Results show that SSLs, previously observed mainly in the Levantine basin and the Adriatic Sea, are present across the whole Mediterranean. There is a marked seasonal variability in the number of SSLs, with a peak in October, and important interannual variability and positive trends in the depth to which they extend across the different regions analyzed.*

*The manuscripts analyze an interesting and yet not very well studied topic. It is well organized and, for the most part, well written. However, there are both formal and content aspects that should be reviewed prior to publication in OS. Overall, I generally agree with Referee #1 in his/her comments, but has also additional suggestions. Therefore, my recommendation is a thorough revision of these points before acceptance.*

*The main aspects to review are as follows:*

*Introduction*

*I strongly agree Referee #1 that that the concept of SSL should be discussed in more depth in the introduction. It is not a well-known concept, as the authors themselves indicate, and should be described and contextualized in the literature so that the reader has a clear idea of its characteristics and relevance.*

We modified the Introduction and added more content on SSL accordingly.

*Method*

*L75-85: Please indicate the dataset from which the Argo profiles are downloaded, dates of the first and last profiles and last access to the dataset.*

We added this information in the updated version.

*L90-95: Please elaborate in the description of the SSI. Since is one of the metrics used to characterize the SSLs, it is important for the reader to understand how it is computed and what represents. You state that is an indicator of the stratification of the water column but a clear idea of how is this representation achieved would be desirable.*

We added the equation and explained it in the text.

*All figures: whenever possible to maintain the same scale on the axes for the different sub-basins, in order to make the comparison of results easier. For example, in Figure 2 the salinity scale varies from one region to another. This is true for other figures and should be avoided as far as possible.
Also in general, keep the color assignment of the lines for the sub-basins from one figure to another. In Figure 1 the blue line corresponds to the Levantine basin, in Figure 5 to the Aegean Sea, in Figure 6 to the Adriatic, etc. Maintain a unified criterion to make it easier to follow the description.
Figure 1: please indicate with lines the specific division of the different sub-basins, particularly between the Ionian and the Levantine Basin, but also for the rest of them.*

We added dashed white-grey lines between different basins.  We also uniformed the colors and basins throughout the whole manuscript and also paid attention to the scales. Thank you for having pointed this out, now it's much clearer.

*Figure 3: what do the dashed vertical white lines in 2021 mean? Please include it in the figure caption.*

The white lines are gaps in the data as the diagram is built as a scatter plot of profiles over time. The explanation is added in the caption.

*Figure 6: What do the dots and bars indicate? I understand the median and extreme values. This should be indicated in the figure caption please.*

We included the description in the caption.

*L109-L112: please rephrase, it is not clear what you mean here.*

We rewrote the sentence.

*L176: '…middle Aegean' -> central Aegean.*

*L185: 'seen' -> observed?*

*L190: 'Levantine Sea' -> Levantine Basin*

*L191: If I understand correctly, higher SSI indicate higher stratification, so the relation between a high stratification and deeper SSLs is not straightforward. I understand that the SSI is computed at the bottom of the SSL, so what you are saying is that a deeper SSLs has a higher SSI at the bottom. It is important to clarify this relationship to understand the results presented in this section.*

Indeed, the relationship is not so straightforward. SSI is dependant on both the density and the SSLs depth (zSG), as seen from the equation (which was added in the manuscript):

$$SSI = g \sum_{i=z_0}^{z_{SG}} (z_i - z_g)(\rho_z - \rho_i)\Delta z_i$$

Therefore, even though an increase in zSG could be present, the density decrease could be the prevailing mechanism for particular months. This could be due to a higher precipitation rate, riverine discharge, or different water masses advection, especially during winter, as it was seen in the Figure 6. Such a surface dilution could bring density and therefore SSI values down, even though the zSG might be deep enough to avoid wintertime convection. We added an explanation in the updated version of the manuscript.

*L193: Cite figure 9.*

We incorporated all the suggestions in the text.

*Discussion*

*L205-L207: It is not clear what you mean here, please rephrased it.*
We rewrote the sentence.

*L208: 'great deal of' -> large?*
*L213: If the Mediterranean Sea is more arid then it either has more freshwater output or less freshwater input. Clarify this here please.*
We rewrote the sentence.

*L117: I agree with Referee #1 that the role of the atmospheric forcing should be discussed more in detail.*

Thank you for having raised this point, which is important to discuss when talking about SSLs trends.
We downloaded ERA5 hourly reanalysis data between 2002 and 2024 for the Levantine region (latitudes between 33 and 36 degrees and longitudes between 22 and 33 degrees).
We obtained winter (January-February-March) daily anomalies for 2024, subtracting the 2002-2024 baseline, for 4 components: latent, sensible, shortwave and longwave fluxes, as well as calculated their sum.
This resulted in negative total anomalies in January and beginning of February, implying surface cooling that in turn causes vertical convection to occur.
We added this figure (see next page) and also an explanation in the text.

[Figure]

*L254-L259: too long sentence, please rephrase.*

*L260: The Mediterranean outflowing waters contribute to a higher salinity of intermediate waters at very specific regions of the North Atlantic. Assuming that it is responsible for the higher salinity of the whole Atlantic is not correct in my opinion. In any case, a reference is missing here.*

Thanks for pointing this out. We rephrased this sentence in order to make it more accurate, as well as added references.

*L272: 'places' -> regions?*

*L275-L279: Here again a deeper analysis of the observed trends would be necessary. You observe a weakening in the vertical gradients of temperature, salinity and density, and at the same time an increase in the SSI, i.e., the stratification. How this two results are compatible?*

As written in the draft, water masses under the SSL depth (zSG) might get saltier and warmer, as it was shown for example in Schroeder et al. (2017) with the warming and salinification of LIW in the Sicily Channel. This in turn deepens the SSL depth (zSG).

SSI in turn might also increase because of the zSG layer increase, as you can see from the equation (that was also added in this version of the manuscript draft and added in the reply above).

With floats we observed the warming and salinification trends by looking at Hovmöller diagrams and seeing strong increases in both T and S at intermediate depths for example in the Levantine Basin, the Adriatic and Ionian seas.

However, as we also underlined in the text, SSI trends are statistically insignificant ($p > 0.01$), therefore these trends should be treated with caution when linking zSG trends with SSI.

*A warmer and saltier LIW, that is currently being observed and also projected by the models could explain the weakening of the gradients but not the increase in the stratification. Are there other studies that support this conclusion?*

SSI is indeed an indicator of stratification, but not of the intensity/increase/gradients of stratification but of the energy which is necessary to mix a layer to a certain depth. Its calculation includes also the depth of the SSLs, as explained in the reply above. So if the SSLs get deeper, without changing densities, this could also be an indicator of increasing SSI. On the other hand, even though an increase in zSG could be present, the density decrease could be the prevailing mechanism preventing SSI to increase, as seen in some winter months, where SSI was not increasing, while zSG did. As the SSLs are explored for the first time, we have no other studies to refer to, but our own.

*Finally, there is a characteristics SSLs that is not mentioned and would be interesting to understand: their extension. SSLs are identified by specific profiles at specific positions, but no information is obtained about the surface over which they are spread. The authors themselves comment on this when describing the SSI calculation. The impact of SSLs on the overall dynamics of the Mediterranean basin or the sub-basins analyzed depends largely on this extent. If they are very localized phenomena or of the meso-scale order. Is it possible to give an order of magnitude of this extent with the collected data?*
*On the other hand, whenever a profile meets the requirements of the algorithm, a new SSL is identified. But it is possible that two or more profiles that are close to each other are identifying the same SSL, if they are structures that remain identifiable for months.*

*I understand that an in-depth analysis of these aspects is not possible with the available data, but in my opinion they should be explained as far as possible.*

This is an interesting point to address. We decided to show this in the form of maps for a certain month when we have most SSLs and another almost devoid of them. Here we attached figures for two years, 2016 and 2021, and for two different months per each of the examined years: one when we have few SSLs (April) and the other when we have the highest percentage of SSLs overall (September/October). Blue dots show profiles where we don't have SSLs, whereas red dots are the ones with SSLs for that specific month and year.

In the first version of the manuscript we already commented the overall highest percentage in the Levantine basin and their ubiquitous presence throughout the entire year, albeit with a smaller number during late winter/ early spring. April months of 2016 and 2021 display separate and spatially isolated cases of SSLs both east and west, seldom being a part of a series of float samplings, as they only start to emerge with the water column stratification (unless deep enough to have resisted the winter convective mixing). Looking at the two figures in September 2016 and 2021, there is clearly a wider spatial extent and a series of samplings that all correspond to SSLs, hence there is also a wider spatial distribution, often resulting in longer trajectories.

We added the new figure (below) and the appropriate explanations in the manuscript.

[Figure]

---

## Referee Report (RR1)

This is my second review of the manuscript. I would like to thank the authors for addressing my previous concerns and suggestions which, in my opinion, greatly improved the manuscript. The findings of the study are better pointed out and clearly brought forward. The revised introduction and discussion further help to set the stage for this study and to place its results within the current state of research on this topic. With only two comments and suggestions left, I now recommend this revised version for publication. Congratulations!

Minor comments:
L30: "With increasing salinity [add reference]"
You should add a citation after this phrase to support the statement

L91-92: 67.554, 64.809, 2.745
Use commas , instead of periods . as thousands separators